# MSTAC: A Multi-Stage Automated Classification of COVID-19 Chest X-ray Images Using Stacked CNN Models

Thanakorn Phumkuea [1,*,†], Thakerng Wongsirichot [2,*,†], Kasikrit Damkliang [2], Asma Navasakulpong [3] and Jarutas Andritsch [4]

1   College of Digital Science, Prince of Songkla University, Songkhla 90110, Thailand
2   Division of Computational Science, Faculty of Science, Prince of Songkla University, Songkhla 90110, Thailand; kasikrit.d@psu.ac.th
3   Division of Respiratory and Respiratory Critical Care Medicine, Prince of Songkla University, Songkhla 90110, Thailand; navasakulpong@hotmail.com
4   Faculty of Business, Law and Digital Technologies, Solent University, Southampton SO14 0YN, UK; jarutas.andritsch@solent.ac.uk
*   Correspondence: 6310025008@psu.ac.th (T.P.); thakerng.w@psu.ac.th (T.W.)
†   These authors contributed equally to this work.

**Abstract:** This study introduces a Multi-Stage Automated Classification (MSTAC) system for COVID-19 chest X-ray (CXR) images, utilizing stacked Convolutional Neural Network (CNN) models. Suspected COVID-19 patients often undergo CXR imaging, making it valuable for disease classification. The study collected CXR images from public datasets and aimed to differentiate between COVID-19, non-COVID-19, and healthy cases. MSTAC employs two classification stages: the first distinguishes healthy from unhealthy cases, and the second further classifies COVID-19 and non-COVID-19 cases. Compared to a single CNN-Multiclass model, MSTAC demonstrated superior classification performance, achieving 97.30% accuracy and sensitivity. In contrast, the CNN-Multiclass model showed 94.76% accuracy and sensitivity. MSTAC's effectiveness is highlighted in its promising results over the CNN-Multiclass model, suggesting its potential to assist healthcare professionals in efficiently diagnosing COVID-19 cases. The system outperformed similar techniques, emphasizing its accuracy and efficiency in COVID-19 diagnosis. This research underscores MSTAC as a valuable tool in medical image analysis for enhanced disease classification.

**Keywords:** COVID-19; CXR; deep learning; CNN; multiclass model





## 1. Introduction

The Coronavirus Disease 2019 (COVID-19) was formally named Severe Acute Respiratory Syndrome Coronavirus-2 (SARS-CoV-2) by the International Committee on Taxonomy of Viruses [1]. According to the primary studies, SARS-CoV-2 was originally transmitted from bats to humans by unknown intermediate animals [2,3]. Most patients exhibit symptoms such as fever, dry cough, headache, chest pain, and shortness of breath [4,5], even though some patients may experience asymptomatic COVID-19 [6,7]. For diagnosing COVID-19, a throat swab is the gold standard [8–10]. Typically, a chest X-ray (CXR) is essential for high-risk patients or Patients Under Inspection (PUI) to detect lung infection. The CXR examination is widely considered the gold standard for diagnosing pneumonia [11]. Specifically, the severity of COVID-19 is exacerbated when patients have risk factors such as older age, high blood pressure, and chronic respiratory disease [12]. The causes of pneumonia include viruses, bacteria, and fungi [13].

Deep learning (DL) serves as an advanced tool for image analysis, object classification, and pattern recognition. DL models accurately identify and classify objects within images, automatically learning complex patterns and representations directly from raw input data. Moreover, they can differentiate patterns and features that are challenging to extract

from large datasets. Consequently, these models can be leveraged to develop computer-aided diagnosis (CADx) systems to effectively analyze medical information. With the assistance of artificial intelligence algorithms, CADx systems enable physicians to provide valuable information and support to fellow healthcare professionals. Furthermore, CADx reduces the workload on them by efficiently analyzing a large amount of medical data. However, it is essential to note that CADx systems are designed to assist rather than substitute for physicians. Final diagnostic decisions should always be made by qualified healthcare professionals, drawing on their clinical knowledge, patient history, and other relevant information.

According to previous studies, CADx may help clinicians identify COVID-19 pneumonia using CXR images [14,15]. Currently, CADx of medical images is becoming increasingly viable. State-of-the-art DL approaches have shown promising results in identifying CXR images of COVID-19 infections [16–18]. Chowdhury et al. [19] proposed a transfer learning approach with binary classification: normal and COVID-19, and three-class classification: normal, viral pneumonia, and COVID-19. The binary classification achieved 99.7% accuracy and sensitivity, while the three-class classification obtained 97.4% accuracy, sensitivity, and F-measure. Afshar et al. [20] proposed the COVID-CAPS modeling framework based on capsule networks, which achieved accuracy and sensitivity of 95.7% and 90.0%, respectively. Wang et al. [21] introduced COVID-net, a deep convolutional neural networks (CNN) design tailored to classify COVID-19, non-COVID-19, and healthy. COVID-net achieved an accuracy of 93.3% and a sensitivity of 93.3%. Ahmad et al. [22] proposed a hybrid ensemble model for identifying COVID-19 infections in CXR images. MobileNet and InceptionV3 models were fine-tuned for the learned weights. The authors performed four-fold cross-validation. The results showed accuracy, precision, and specificity of 94.2%, 89.9%, and 88.3%, respectively.

Furthermore, Khuzani, Heidari, and Shariati [23] used a dimensionality reduction method to generate a set of optimal features of CXR images to develop a COVID-classifier that distinguishes COVID-19 cases from non-COVID-19 cases. The proposed COVID-classifier achieved an accuracy of 94.05%. Yang et al. [24] used VGG16 to detect multiclass COVID-19 CXR and CT-scan images. The accuracy of the three-class classification of CXR images was 97%. Hong et al. [25] proposed a lightweight CNN, MGMADS-CNN that classifies COVID-19 CXR images with an accuracy of 96.75%. Nishio et al. [26] developed EfficientNet to classify COVID-19 pneumonia, non-COVID-19 pneumonia, and healthy CXR images. Three-class classification accuracy was 86.67%. According to Table 1, DL and pre-trained models for COVID-19 detection on CXR images dominated this research field. However, several studies used small and unbalanced datasets to develop models. This raises potential problems with generalization and bias. Small datasets may restrict the model's ability to understand the full complexity of cases, and unbalanced datasets can result in biased performance towards the majority class. Even with the prevalence of CXR-pretrained models, researchers should be careful about the size and balance of their datasets to ensure strong and unbiased model performance in detecting COVID-19.

Our retrospective study mainly aimed to classify COVID-19, non-COVID-19, and healthy CXR images using a stacked CNN model. Using stacked CNN models, multi-stage automated classification (MSTAC) of COVID-19 CXR images was realized based on two CNN-based binary classification models. The first model differentiates healthy and unhealthy CXR images. Consequently, all the unhealthy CXR images are passed to the second CNN-based binary classification model. The second model mainly classifies COVID-19 and non-COVID-19 CXR images. We compared MSTAC with the CNN model that classifies COVID-19 pneumonia, non-COVID-19, and healthy. However, only a few studies have used multi-stage automated detection for classifying COVID-19 based on CXR images.

**Table 1.** Summary of previous studies on COVID-19 CXR detection.

| Author | Methods | Classes | Dataset |
|---|---|---|---|
| Chowdhury et al. [19] | Different pre-trained CNN model | 3 | COVID-19 = 423, Viral pneumonia = 1485, and Normal = 1579 |
| Afshar et al. [20] | COVID-CAPS framework consists of 4 convolutional layers and 3 capsule layers | 4 | Not specified |
| Wang et al. [21] | COVID-Net network architecture | 3 | 13,975 CXR images |
| Ahmad et al. [22] | Hybrid ensemble model | 4 | COVID-19 = 1050, Viral pneumonia = 1050, Bacterial pneumonia = 1050, and Normal = 1050 |
| Khuzani et al. [23] | Dimensionality reduction method of CXR images to build classifier | 3 | COVID-19 = 140, non-COVID-19 = 140, and Normal = 140 |
| Yang et al. [24] | VGG16 | 3 | COVID-19 = 3616, Pneumonia = 1345, and Normal = 3500 |
| Hong et al. [25] | MGMADS-CNN model | 4 | COVID-19 = 1341, Viral pneumonia = 1733, Bacterial pneumonia = 2705, and Normal = 2916 |
| Nishio et al. [26] | Several transfer learning techniques | 3 | Three datasets |
| Our MSTAC | Stacked CNN-based | 3 | COVID-19 = 1050, non-COVID-19 = 1050, and Normal = 1050 |

The following are the main contributions of our study that aim to address these issues:

(1) We propose the Multi-Stage Automated Classification (MSTAC) for COVID-19 detection using two stacked CNN-based binary classification models and a large dataset of CXR images.

(2) The first model for MSTAC (stage-1) is a DL model that classifies between healthy and unhealthy CXR images.

(3) The second model for MSTAC (stage-2) is a DL model that classifies between COVID-19 and non-COVID-19 CXR images.

(4) The proposed MSTAC model stacks stage-1 and stage-2. This stacking allows MSTAC to identify three classes of CXR images: COVID-19, non-COVID-19, and healthy.

(5) MSTAC achieves high accuracy in identifying COVID-19 pneumonia cases from CXR images. This proposed model enhances the efficiency and accuracy of COVID-19 detection.

(6) We also compare MSTAC with previous studies that classify COVID-19 pneumonia, non-COVID-19, and healthy cases. This comparison demonstrates the promising results of MSTAC in COVID-19 classification and its effectiveness for medical image analysis and diagnosis.

Overall, the MSTAC for COVID-19 model aims to make a precise contribution to the ongoing struggle with the COVID-19 pandemic.

The study's structure is organized as follows: The methodology and proposed models are discussed in Section 2. The experimental findings are presented in Section 3. In Section 4, discussions are provided. Finally, Section 5 presents the conclusion.

## 2. Materials and Methods

### 2.1. Dataset

Three publicly available data repositories were used: (1) CXR images (pneumonia) dataset available on Mendeley data, which contains CXR images of healthy and non-COVID-19 pneumonia [27]; (2) augmented COVID-19 X-ray images dataset [28]; and (3) COVID-19 image data [29] for CXR images of COVID-19 pneumonia. The representative CXR images of healthy, non-COVID-19 pneumonia, and COVID-19 in the combined dataset are shown in Figure 1.

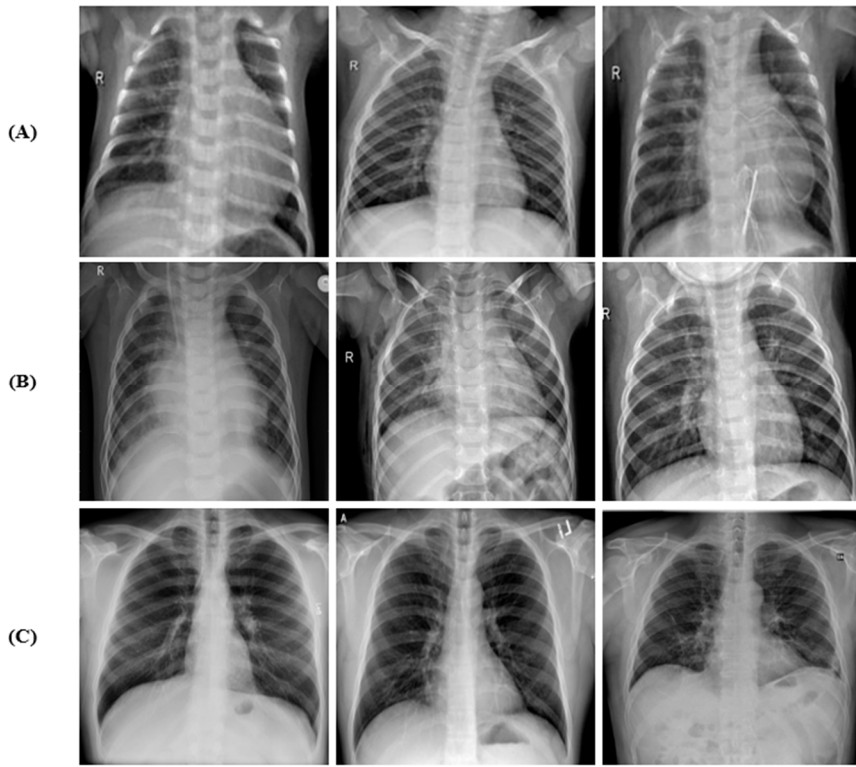

**Figure 1.** Sample of CXR images: (**A**) healthy, (**B**) non-COVID-19, and (**C**) COVID-19.

The dataset used comprised 3150 CXR images. Specifically, 1050 images were randomly selected from dataset (1) for both the healthy and non-COVID-19 pneumonia classes. For COVID-19 pneumonia, 912 CXR images were collected from dataset (2). A total of 138 CXR images were randomly selected from dataset (3) to prevent a class imbalance. Our dataset included 3150 CXR images from all three classes, as shown in Table 2, and was used to develop and validate our proposal.

**Table 2.** Dataset distribution of CXR images.

| Dataset | Classes | Number of Images |
|---|---|---|
| CXR Images (Pneumonia) dataset [27] | Healthy | 1050 |
| CXR Images (Pneumonia) dataset [27] | Non-COVID-19 | 1050 |
| Augmented COVID-19 X-ray Images Dataset [28] | COVID-19 | 912 |
| COVID-19 Image Data Collection [29] | COVID-19 | 138 |

*2.2. DL Architecture*

The DL architecture employed in our study utilizes a sequential CNN for the classification of CXR images. As the CXR images were gathered from various public sources, it is necessary to standardize the image size. To ensure uniformity, the CXR images were standardized to dimensions of 224 × 224. Scaling CXR images to 224 × 224, despite their relatively low resolution, this study was conducted under the influence of resource constraints. Despite the reduced resolution, these previous studies have consistently demonstrated the feasibility of developing highly efficient models for tasks such as classification. The initial batch normalization layer played a crucial role in normalizing inputs, significantly reducing the required training epochs for deep networks. The sequential CNN includes three convolutional layers (Conv) of 3 × 3 size. There are different sizes of Conv filters, the first two layers have 64 filters, and the third has 32 filters. Next to each Conv is a 3 × 3 max pooling layer. The third Conv's pooling output is fed to the first fully connected layer,

which includes 128 neurons. A dropout of 0.35 was applied in the second Conv, the third Conv, and the fully connected layers. We used the Adam optimizer to update weights and perform the cross-entropy loss function. The CNN model was trained using 20 epochs with a batch size of 32. The architecture of the proposed CNN model is shown in Figure 2.

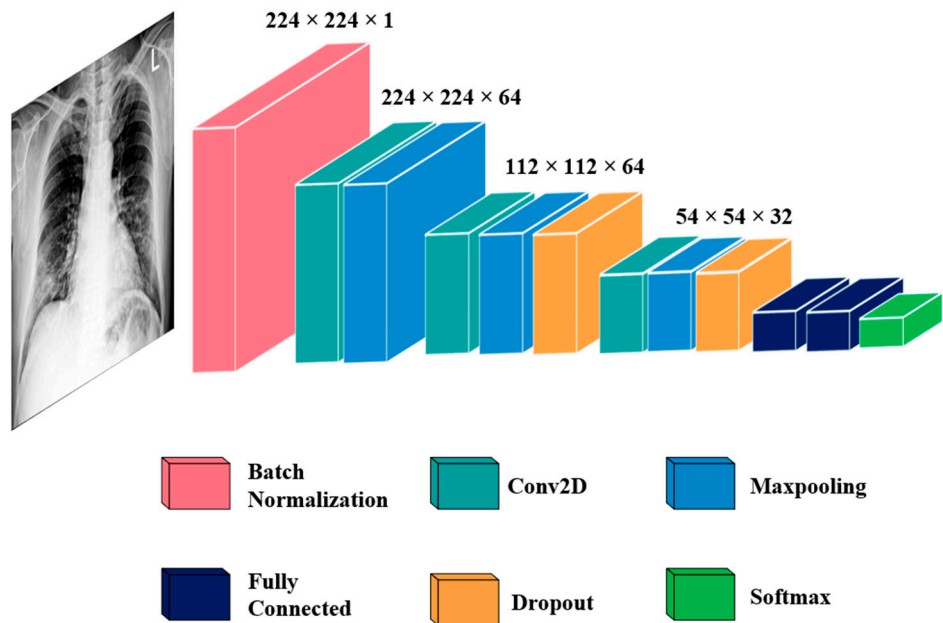

**Figure 2.** Architecture of the CNN model.

### 2.3. Multi-Stage Automated Classification (MSTAC)

The MSTAC stacked multiple binary classification models to classify COVID-19, non-COVID-19, and healthy CXR images. The model stacking approach was inspired by the stacked machine learning model used to classify sleep stages automatically [30]. In this study, MSTAC was designed to have two stages. The stage-1 model performs the classification of healthy CXR images from unhealthy ones. The stage-2 model classifies between COVID-19 and non-COVID-19 CXR images. Figure 3. illustrates the block diagram of our proposed experiments and the evaluation of MSTAC. From the block diagram, the CXR images were diagnosed as unhealthy by the stage-1 model. After that, unhealthy images were diagnosed by the stage-2 model to determine whether they were COVID-19 or non-COVID-19. The development and test set were used to develop and evaluate classification model performance. We randomly split the CXR image dataset into training and test sets at an 80:20 ratio. The test set had 630 CXR images, comprising COVID-19 pneumonia, non-COVID-19 pneumonia, and normal.

To elaborate, in the initial step, the stage-1 model successfully classified 630 CXR images into categories of healthy and unhealthy. Upon completion of the stage-1 predictions, the algorithm automatically stored the images predicted as belonging to the healthy class. Subsequently, the stage-2 model took charge of classifying the unhealthy class images to discern whether they were indicative of COVID-19 or non-COVID-19 conditions. The predictions generated by the stage-2 model were then saved utilizing the same algorithm. Finally, we employed the recorded predictions, stored by the algorithm, to conduct a comparative analysis with the true labels of the CXR images. This facilitated the creation of a confusion matrix, serving as a robust tool to evaluate the overall performance of MSTAC.

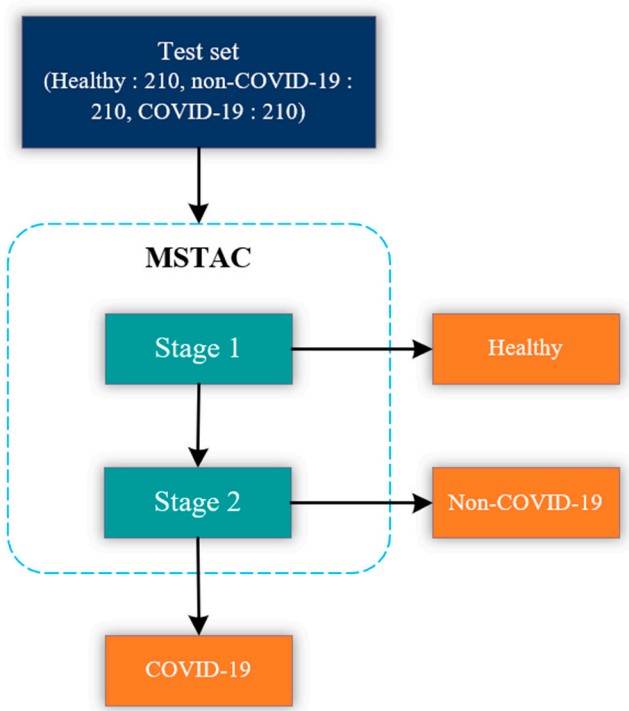

**Figure 3.** Illustration of MSTAC.

For the stage-1 model, we used the training set by combining the COVID-19 class with the non-COVID-19 class. Considering the design, this process imbalanced the classes and caused bias in the evaluation. We applied the synthetic minority oversampling technique (SMOTE) to solve the imbalanced classes problem [31]. SMOTE synthesized new data from existing data using k-nearest neighbors (kNN) and inserted them into the original dataset [32]. Originally, there were 840 subjects in the healthy class and 1680 in the unhealthy class. The healthy class became a minority in the training set for the stage-1 model. Consequently, we synthesized 840 healthy images using SMOTE to balance both classes. The stage-1 model was finally performed with 1680 CXR healthy and 1680 CXR unhealthy images. To build the stage-2 model, we eliminated the healthy class from the training set. The stage-2 model was trained using 840 COVID-19 and 840 non-COVID-19 CXR images. Table 3. summarizes the CXR images used for developing the CNN model. The use of SMOTE was intended to address the effects of class imbalance. This ensures that the CNN model of the stage-1 could effectively learn and make accurate predictions for both the healthy and unhealthy classes in the CXR images.

**Table 3.** CXR image distribution used to develop each CNN model.

| CNN-Based | Classes | No. of Images in Training Set (80%) | |
| --- | --- | --- | --- |
| | | Original | SMOTE |
| stage-1 | Healthy | 840 | 1680 |
| | Unhealthy | 1680 | 1680 |
| stage-2 | COVID-19 | 840 | - |
| | Non-COVID-19 | 840 | - |

### 2.4. CNN-Multiclass

In our comparison, we also made a separate deep model named CNN-Multiclass to classify CXR images into COVID-19, non-COVID-19, and healthy classes. The architecture of CNN-Multiclass mirrors that of MSTAC's CNN model, with a singular modification—expanding the input dimension from 2 to 3 to accommodate the multiclass

prediction task. This adjustment ensures that CNN-Multiclass aligns with the structure of MSTAC, enabling us to assess and compare the classification performance of each model for specific health conditions in CXR images.

CNN-Multiclass was trained using 840 CXR images for each of the three categories: COVID-19, non-COVID-19, and healthy, as detailed in Table 4. To maintain consistency in the evaluation process, we employed the same test set used for MSTAC to assess CNN-Multiclass. This test set consisted of 630 CXR images, covering cases of COVID-19, non-COVID-19, and healthy conditions. This test set of images served as the standardized benchmark for evaluating the performance of both CNN-Multiclass and MSTAC, as shown in the diagram in Figure 4.

**Table 4.** CXR image distribution used to develop CNN-Multiclass model.

| Model | Classes | No. of Images in Training Set (80%) |
|---|---|---|
| CNN-based | Healthy | 840 |
| | Non-COVID-19 | 840 |
| | COVID-19 | 840 |

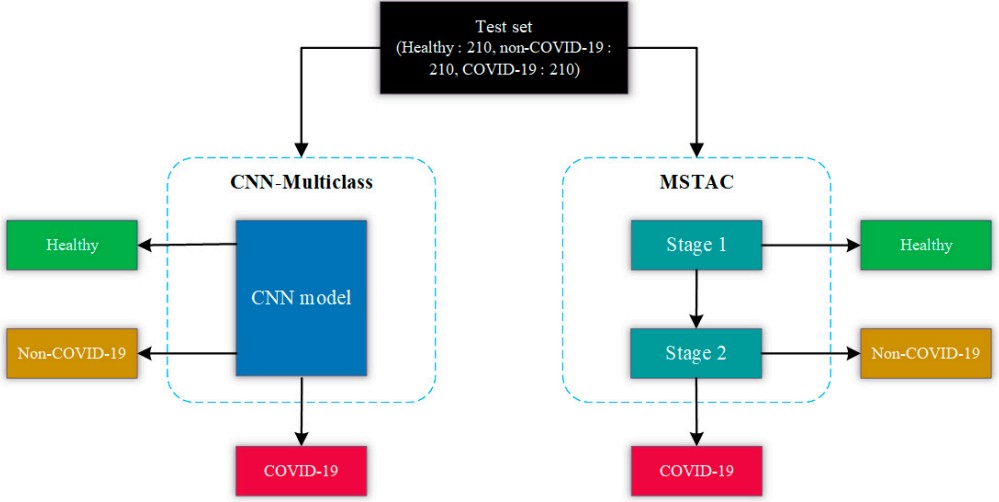

**Figure 4.** Illustration of Performance Evaluation: CNN-Multi Class vs. MSTSC.

By using the identical test set, potential biases were minimized. This ensures a fair comparison of how well each model can accurately classify and distinguish among COVID-19, non-COVID-19 pneumonia, and normal CXR images. This approach enhances the reliability of the evaluation results. It provides a robust basis for comparing the effectiveness of the two models in the context of their diagnostic performance on standardized data.

## 3. Results

In this section, we present the experimental results to demonstrate the ability of the MSTAC model to identify COVID-19 instances from CXR images.

### 3.1. Performance Evaluation

The evaluation included several performance measures: accuracy (*ACC*), sensitivity (*TPR*), precision (*PPV*), F1-score (*F1*), and the area under the curve (*AUC*) [33]. *ACC* gauges the overall correctness of classifications, indicating the proportion of accurately classified cases among all assessed cases. *TPR* quantifies the proportion of correctly identified positive cases among all actual positive cases, signaling low false negatives when high. *PPV* measures the proportion of true positive cases among all cases classified as positive by the system, highlighting low false positives when high. *F1* strikes a balance between

sensitivity and precision, offering a single metric to assess both true positives and false positives. *AUC* reflects the system's overall discriminative power in distinguishing between different classes, particularly crucial for imbalanced datasets.

The performance measure formulae are presented in Equations (1)–(5), involving true positive (*TP*), true negative (*TN*), false positive (*FP*), and false negative (*FN*). Additionally, we employed the area under the receiver operating characteristic curve (AUC of ROC) to assess performance. The DL algorithms were implemented using Python 3.8.13 with TensorFlow 2.4, CUDA 11.0, cuDNN v8.0.5 on an NVIDIA GeForce-1050 GPU (Nvidia, Santa Clara, CA, USA) with 4 GB GDDR5 onboard memory. The system ran on an Intel Core i3-8100 3.6 GHz processor (Intel, Santa Clara, CA, USA), paired with 8 GB DDR4 RAM.

$$ACC = \frac{TP + TN}{(TP + FP) + (TN + FN)} \tag{1}$$

$$TPR = \frac{TP}{(TP + FN)} \tag{2}$$

$$PPV = \frac{TP}{(TP + FP)} \tag{3}$$

$$F1 = \frac{2TP}{(2TP + FP + FN)} \tag{4}$$

$$AUC = \frac{1}{2}\left(\frac{TP}{TP + FN} + \frac{TN}{TN + FP}\right) \tag{5}$$

### 3.2. Performance Results

Figure 5 shows the training and validation performance of the stage-1 model. The stage-1 model was trained with 20 epochs to classify healthy and unhealthy CXR images. The training accuracy was stable at 99.2% after epoch 15. The training and validation losses were similar, representing a good fit for the stage-1 model, and were not overfitted. In Figure 6, the accuracy plot of the stage-2 model shows a good training and validation trend. Despite a slight difference in the gap between the training and validation curves in both accuracy and loss plots, this indicates that the stage-2 model is effective. It is effective in distinguishing between COVID-19 and non-COVID-19 CXR images. The results of the combined stage-1 and stage-2 models are showcased in MSTAC, as depicted in Figure 3. To assess the performance of MSTAC and CNN-Multiclass, a set of 630 CXR images was utilized. The training and validation results, displayed in the figures, suggest the efficacy of the proposed multi-stage approach in classifying different health conditions in CXR images.

Table 5 compares the outcomes of the three-class classification for CNN-Multiclass and MSTAC. The test set comprised 210 CXR images for each of the three classes that had not been part of the training or testing sets. Initially, MSTAC attained accuracy, precision, recall, f1-score, and AUC values of 97.30%, 97.34%, 97.30%, 97.30%, and 97.98%, respectively. MSTAC showed a three-class classification result that was better than CNN-Multiclass in terms of effectiveness. Figure 7 displays the confusion matrix, showcasing robust sensitivity for COVID-19 and non-COVID-19 pneumonia. In confusion matrix (a), CNN-Multiclass accurately classified 207 out of 210 COVID-19 cases and 208 out of 210 non-COVID-19 cases, achieving sensitivities of 98.57% and 99.05%, respectively. In confusion matrix (b), MSTAC correctly classified 208 out of 210 COVID-19 pneumonia cases and 206 out of 210 non-COVID-19 pneumonia cases, with sensitivities of 99.05% and 98.10%, respectively. However, this suggests that the accuracy in classifying COVID-19 and non-COVID-19 was higher than that for the healthy class. Nevertheless, the high sensitivity of COVID-19 detection can reduce the burden on the healthcare system.

The ROC curves displayed in Figure 8a,b offer a comprehensive overview of the classification performance of our proposed MSTAC and the CNN-Multiclass model. For CNN-Multiclass, the Area Under the Curve (AUC) values for healthy, non-COVID-19, and

COVID-19 were 0.930, 0.961, and 0.992, respectively. These values collectively resulted in an overall AUC of 0.961 for the three-class classification. The AUC of MSTAC for healthy, non-COVID-19, and COVID-19 was 0.969, 0.9774, and 0.993, respectively. This represents 0.980 of the classification. Consequently, the findings suggest that our proposed MSTAC outperforms CNN-Multiclass in terms of overall performance, specifically in COVID-19 detection and AUC-ROC.

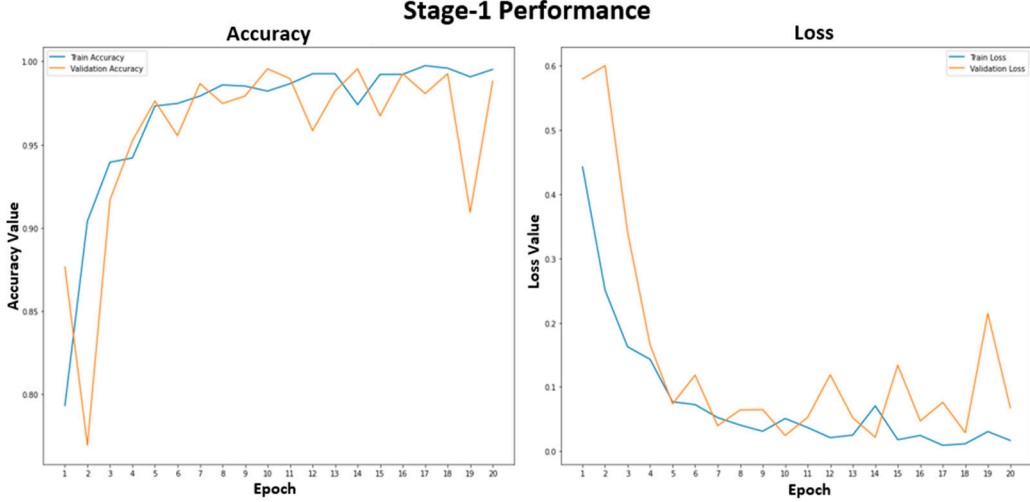

**Figure 5.** Stage-1 model performance for the healthy and unhealthy CXR images.

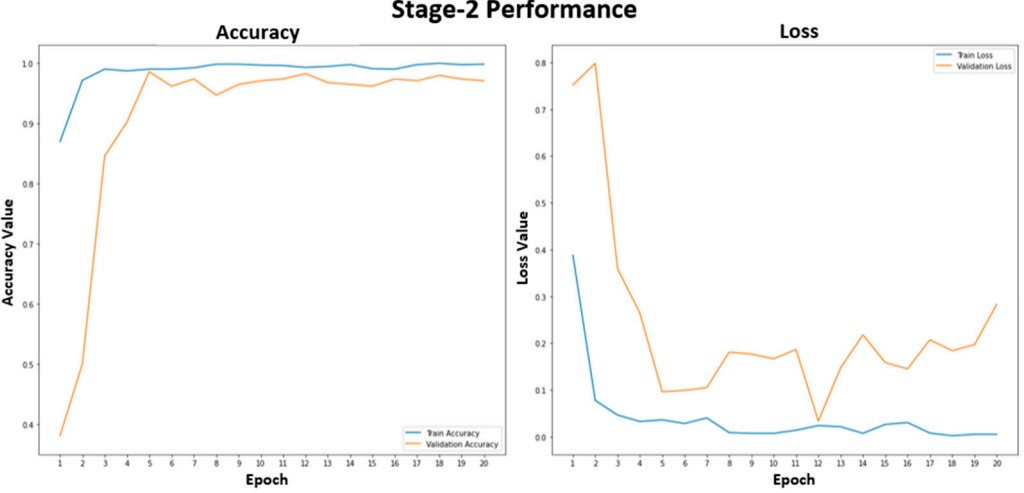

**Figure 6.** Stage-2 model performance for the COVID-19 and non-COVID-19 CXR images.

**Table 5.** Classification performance comparison in classifying the three-class CXR images. The best results are shown in bold.

| Three-Classes Classification | Performance Metric (%) | | | |
|---|---|---|---|---|
| | Accuracy | Precision | Sensitivity | F1-Score |
| CNN-Multiclass | 94.76 | 95.22 | 94.76 | 94.75 |
| MSTAC | **97.30** | **97.34** | **97.30** | **97.30** |

Bold characters indicate higher values.

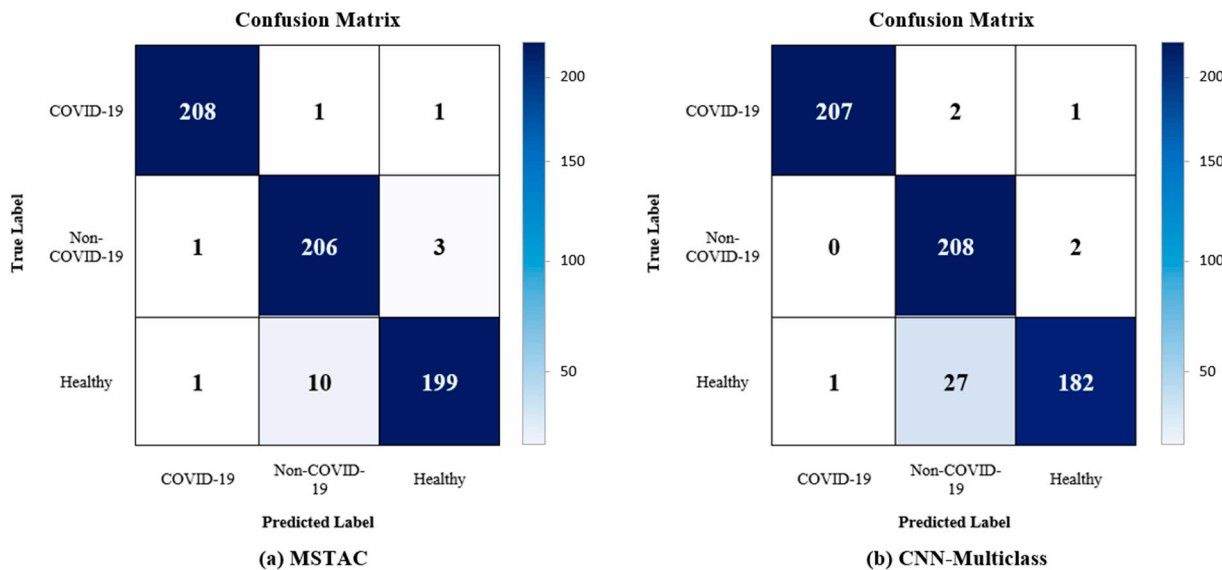

**Figure 7.** Confusion matrix for the three-class classification (**a**) Confusion matrix of the MSTAC (**b**) Confusion matrix of the CNN-Multiclass.

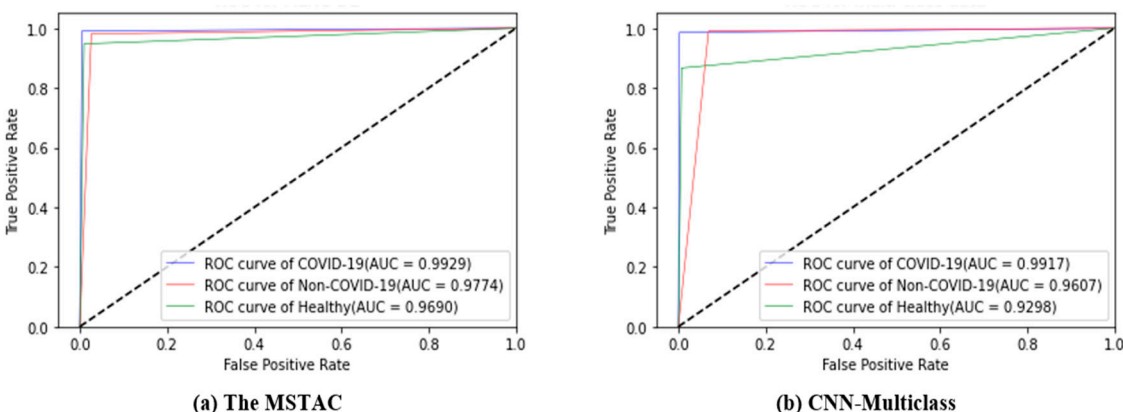

**Figure 8.** ROC curves of different classifications: (**a**) MSTAC; (**b**) CNN-Multiclass.

## 4. Discussion

This retrospective study represents a significant advance in the development of MSTAC, a model designed for the challenging task of three-class classification—distinguishing between COVID-19 pneumonia, non-COVID-19 pneumonia, and healthy cases through the analysis of CXR images. The utilization of stacked multiple binary models was a detailed process, demanding considerable effort in the reconstruction of the development set and the subsequent training of each model. The proposed CNN exhibited efficiency, converging in 20 epochs, each lasting approximately 18 s. The paramount outcome of our investigation is the noteworthy diagnostic accuracy achieved by MSTAC, surpassing the 97% threshold across the three distinctive classes. In the field of deep learning, numerous studies have investigated a range of preprocessing techniques, feature extraction methodologies, and classification approaches. Notably, customized CNNs and ensemble learning have emerged as pivotal strategies. Given the gravity of potential risks associated with misdiagnoses, our primary objective was to craft a model that excels in accuracy. Table 6 compares the performances achieved in compared studies. With an accuracy exceeding 95%, our model establishes itself as a statistical benchmark. We observed that our proposed approach of stacking multiple binary models achieved better performance than that achieved in studies that used a similar number of CXR image datasets. Comparatively, the performance of MSTAC places it among the top-performing models in the field. It outperforms the accuracy

achieved in several notable studies including Wang et al. [21], Khuzani et al. [23], Yang et al. [24], and Nishio et al. [26] in terms of three-class classification. Another strength of MSTAC is its competitive sensitivity of 97.30%, indicating its effectiveness in correctly identifying true positive cases across all three classes. This sensitivity outperforms several existing models, highlighting MSTAC's robustness in detecting COVID-19, non-COVID-19 pneumonia, and healthy CXR images. Compared to the various transfer learning techniques utilized by Yang et al. [24] and Nishio et al. [26], MSTAC demonstrates notably higher performance. This underscores the effectiveness of our approach in achieving superior results. Additionally, the robustness of the MSTAC model is significantly influenced by the dataset size. The use of a well-balanced dataset, comprising 1050 images for each class and totaling 3150 CXR images, contributes to a more comprehensive and diverse set for both training and evaluation. This stands in clear contrast to studies employing smaller datasets, as shown by Khuzani et al. [23]. It emphasizes MSTAC's advantage in efficiently addressing diverse features and enhancing its overall effectiveness in detecting COVID-19.

**Table 6.** Comparison of classification results of our MSTAC with other COVID-19 DL studies on CXR images.

| Author | Methods | Classes | Dataset | Accuracy (%) | Sensitivity (%) |
|---|---|---|---|---|---|
| Chowdhury et al. [19] | Different pre-trained CNN model | 3 | COVID-19 = 423, Viral pneumonia = 1485, and Normal = 1579 | 97.9 | 97.9 |
| Afshar et al. [20] | COVID-CAPS framework consists of 4 convolutional layers and 3 capsule layers | 4 | Not specified | 95.7 | 90 |
| Wang et al. [21] | COVID-Net network architecture | 3 | 13,975 CXR images | 93.3 | 93.3 |
| Ahmad et al. [22] | Hybrid ensemble model | 4 | COVID-19 = 1050, Viral pneumonia = 1050, Bacterial pneumonia = 1050, and Normal = 1050 | 94.2 | 88.4 |
| Khuzani et al. [23] | Dimensionality reduction method of CXR images to build classifier | 3 | COVID-19 = 140, non-COVID-19 = 140, and Normal = 140 | 94.05 | - |
| Yang et al. [24] | VGG16 | 3 | COVID-19 = 3616, Pneumonia = 1345, and Normal = 3500 | 97 | 97 |
| Hong et al. [25] | MGMADS-CNN model. | 4 | COVID-19 = 1341, Viral pneumonia = 1733, Bacterial pneumonia = 2705, and Normal = 2916 | 96.75 | 96.60 |
| Nishio et al. [26] | Several transfer learning techniques | 3 | Three datasets | 86.76 | - |
| MSTAC (our proposed model) | Stacked CNN-based | 3 | COVID-19 = 1050, non-COVID-19 = 1050, and Normal = 1050 | 97.30 | 97.30 |

The MSTAC architecture is intentionally designed for optimal performance in detecting COVID-19. The two-stage classification process initially categorizes cases into healthy and unhealthy, followed by a precise classification into COVID-19 and non-COVID-19 categories. The stacking approach integrates the strengths of individual models. This allows for a comprehensive analysis of the three distinct classes within CXR images.

The clinical importance of MSTAC extends beyond its technical strengths. The model's potential for high accuracy and efficiency promises to significantly reduce misdiagnoses, thereby minimizing risks for patients. The efficient diagnostic workflow streamlined by MSTAC shows potential for integration into current healthcare systems. This is especially important in environments where rapid and accurate COVID-19 detection is crucial.

While MSTAC has demonstrated considerable success, it is crucial to acknowledge a limitation in its accuracy for healthy cases when compared to its robust performance in identifying COVID-19 and non-COVID-19 cases. In the future, we could improve the model by looking at more detailed features, using advanced training methods, and maybe including more data sources to make it better at identifying healthy cases. Our approach utilizes existing resources without the need for extensive processing, providing a straightforward and efficient means of detecting COVID-19 CXR images.

Our distinctive stacking model technique, which has played a crucial role in the success of MSTAC, explores a less-traveled path in the field of CXR image analysis for detecting COVID-19. The simplicity and efficiency of our approach allow it to utilize existing resources without the need for extensive processing. These attributes position MSTAC as a promising system in the realm of diagnostic tools for respiratory conditions.

## 5. Conclusions

The MSTAC model offers a promising method for accurately and efficiently detecting COVID-19 from CXR images. By using stacked CNN models and a two-stage classification strategy, MSTAC achieves high diagnostic performance. The system showcases impressive accuracy, sensitivity, precision, F1-score, and AUC. This suggests its potential as an effective tool for COVID-19 diagnosis. Our study makes significant contributions, such as developing a three-class classification system that can distinguish COVID-19, non-COVID-19, and healthy cases. Additionally, we validated MSTAC on publicly available datasets to ensure transparency and reproducibility. Comparisons with traditional CNN models underscore MSTAC's superiority in COVID-19 detection.

In the future, we will implement image data augmentation techniques on CXR images to enhance classification accuracy. Additionally, our current study focused on classifying three categories. However, our future plans involve expanding the dataset and creating a model capable of classifying four classes. This will involve incorporating a new binary classification model for distinguishing between viral and bacterial pneumonia.

Furthermore, our future objectives include developing a classification model that utilizes a pretrained model to identify abnormal regions in CXR images. Lastly, we aim to create a prototype of a web-based CADx system. This system could serve as an alternative, bringing the technique into clinical settings as a valuable tool to support medical staff in improving COVID-19 detection accuracy. At the core of this system is the diagnosis of CXR images uploaded by users, which are classified by MSTAC into healthy, pneumonia, or COVID-19 cases.

**Author Contributions:** Conceptualization, T.P. and T.W.; methodology, T.P. and T.W.; software, T.P.; validation, T.P. and T.W.; formal analysis, T.P. and T.W.; writing—original draft preparation, T.P. and T.W.; writing—review and editing, T.P., T.W., K.D., A.N., and J.A.; supervision, T.W.; funding acquisition, T.P. All authors have read and agreed to the published version of the manuscript.

**Funding:** This research received no external funding.

**Institutional Review Board Statement:** Not applicable.

**Informed Consent Statement:** Not applicable.

**Data Availability Statement:** The dataset utilized in this study is openly accessible to the public. Additionally, all the code developed to support the findings presented in this publication has been made publicly available. The code repository can be accessed at the following URL: https://github.com/tanthanakorn2541/MSTAC_Deep_Learning_COVID19 (accessed on 21 October 2022)".

**Conflicts of Interest:** The authors declare no conflict of interest.

## Abbreviations

MSTAC     Multi-Stage Automated Classification.
CXR         Chest X-Ray.
CNN       Convolutional Neural Networks.
DL           Deep Learning.
CADx      Computer-Aided Diagnosis.

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
