# Peer review of "MSTAC: A Multi-Stage Automated Classification of COVID-19 Chest X-ray Images Using Stacked CNN Models"

_tomography, doi:10.3390/tomography9060173_

Round 1
Reviewer 1 Report
Comments and Suggestions for Authors
This study proposed a multi-stage automated classification model using simple CNNs. I have the following comments on the current manuscript.
1. Figure 2 shows the architecture of the CNN model. I wonder how the two CNN models are stacked. Please replace current figure with the complete stacked CNN models (with inputs and outputs) in the manuscript.
2. For the CRS images in this study, what are the original resolutions (e.g., 1024x1024 pixels)? What was the bit depth per pixel (e.g., 12 bits per pixel)? What was the typical intensity range? Please include these information in the manuscript.
3. In this study, the CXR images were scaled to 224x224, which have a relatively low resolution. How will the image resolution affect classification performance?
4. In Table 4, the best results were not shown in bold. Also, the Accuracy and Sensitivity of MSTAC is 97.3%.
In Table 5, the Accuracy and Sensitivity of Rajaraman et al [19] is 99% (1st row of Table 5), but the introduction section (Line 62-63) says “It achieved the highest accuracy of 97.42% for a three-class classification of CXR images.” Which one the correct number?
Anyway, both results seem higher than MSTAC (97.3%). Could the authors discuss the pros/cons between the proposed method and that in Rajaraman et al [19]?
Please check all the references number because the authors use both Rajaraman et al [17] and Rajaraman et al [19] throughout the manuscript.
Comments on the Quality of English LanguageN/A
Author Response
Reviewer 1
- Figure 2 shows the architecture of the CNN model. I wonder how the two CNN models are stacked. Please replace current figure with the complete stacked CNN models (with inputs and outputs) in the manuscript.
resp: Thank you for your valuable feedback; we employed a simple method to stack 2 models using a specific algorithm. This algorithm was exclusively utilized during the testing process. The MSTAC was evaluated with the test set. Firstly, the stage-1 model classified 630 CXR images into healthy and unhealthy classes. After completing the stage-1 model, the algorithm saved images predicted as healthy by the model. After that unhealthy class was classified by the stage-2 model to determine whether they were COVID or non-COVID. The predictions of the stage-2 model were saved using the same algorithm. Finally, we compared the algorithm-recorded predictions with the true labels, creating a confusion matrix to evaluate MSTAC's performance (Lines 180-189).
- For the CRS images in this study, what are the original resolutions (e.g., 1024x1024 pixels)? What was the bit depth per pixel (e.g., 12 bits per pixel)? What was the typical intensity range? Please include these information in the manuscript.
resp: I have revised and improved it as suggested. (Lines 147-150)
- In this study, the CXR images were scaled to 224x224, which have a relatively low resolution. How will the image resolution affect classification performance?
resp: Due to limited resources, we scaled the images to a similar size to many previous studies. Low image resolution may result in a reduction of image details. However, from many previous studies we have reviewed, it has been observed that despite the low image resolution, it is possible to develop highly efficient models. (Lines 150-154).
- In Table 4, the best results were not shown in bold. Also, the Accuracy and Sensitivity of MSTAC is 97.3%.
resp: I have revised and improved it as suggested (Table 5).
- In Table 5, the Accuracy and Sensitivity of Rajaraman et al [19] is 99% (1strow of Table 5), but the introduction section (Line 62-63) says “It achieved the highest accuracy of 97.42% for a three-class classification of CXR images.” Which one the correct number?
Anyway, both results seem higher than MSTAC (97.3%). Could the authors discuss the pros/cons between the proposed method and that in Rajaraman et al [19]?
Please check all the references number because the authors use both Rajaraman et al [17] and Rajaraman et al [19] throughout the manuscript.
resp: To solve the confusing problem, we carefully considered excluding Rajaraman's research from the manuscript.
Reviewer 2 Report
Comments and Suggestions for Authors
This manuscript proposed a multi-stage automated classification (MSTAC) of COVID 19 CXR images using stacked CNN models to differentiate COVID-19, non-COVID-19, and healthy cases. Three public datasets were used for training and testing. Their results suggested that the overall classification showed better accuracy and sensitivity compared with other techniques. The topic of the manuscript has been specially designed, the manuscript has been well written, and has good contributions in their research fields.
I have only one suggestion:
Authors should pay more attention to the format of the texts. For example, L-90, L-212-220, Table 4, Table 5.
Comments on the Quality of English LanguageThe quality of English Language is good.
Author Response
Reviewer 2
This manuscript proposed a multi-stage automated classification (MSTAC) of COVID 19 CXR images using stacked CNN models to differentiate COVID-19, non-COVID-19, and healthy cases. Three public datasets were used for training and testing. Their results suggested that the overall classification showed better accuracy and sensitivity compared with other techniques. The topic of the manuscript has been specially designed, the manuscript has been well written, and has good contributions in their research fields.
I have only one suggestion:
Authors should pay more attention to the format of the texts. For example, L-90, L-212-220, Table 4, Table 5.
resp: Thank you for your valuable feedback; we have revised and improved the formatting as suggested.
Reviewer 3 Report
Comments and Suggestions for Authors
1. It is important to wrtie down the full name when an abbreviation is presented. It would be nicer if you could explain the full name of VGG as the specific term is presented for the first time in this article (Line 79)
2. From line 57 to line 75, this article lists the accuracy level of each study. However, this list lacks a supporting logic. Did authors list these study results in a chronological order? Alphabetic order. When listing these multiple study results, it would be nicer if authors could clarify how they came up with this listing.
3. Authors focued on the diagnosis of COVID-19 based on CXR. As the most common diagnosis method of COVID-19 is In-Vitro Diagnostics approach such as PCR, authors need to mention the other types of diagnosis for COVID-19. Then authors could explain that the CXR approach of the diagnosis of COVID-19 has a specific advantage or strength over other test methods. This explanation could strengthen the value of this study in a more comprehensive manner.
Comments on the Quality of English LanguageThis article has many grammatical errors. Though I can't point out all, I believe that these errors need to be corrected in a tight manner.
1. (Line 48) "are difficult from" needs to be corrected
2. (Line 51) "support to" needs to be corrected
3. (Line 135) the number "1050" is used twice in the sentence, so it is hard to fully understand what authors want to explain
4. (Line 162) the word "classify" is verb, and noun form is more appropriate in the sentence
5. (Line 170) I would assume that authors wanted to mention "test set" instead of "test rate"
6. (Line 248) I would like to discourage the usage of subjective word such as "remarkable" in this type of scientific article
7. (Line 280) I would like to discourage the usage of subjective word such as "perfect" in this type of scientific article
Author Response
Reviewer 3
This article has many grammatical errors. Though I can't point out all, I believe that these errors need to be corrected in a tight manner.
- (Line 48) "are difficult from” needs to be corrected
resp: Thank you for your valuable feedback; I have revised and improved it as suggested (Lines 48-49).
- (Line 51) "support to" needs to be corrected
resp: I have revised and improved it as suggested (Lines 51-52).
- (Line 135) the number "1050" is used twice in the sentence, so it is hard to fully understand what authors want to explain
resp: I have revised and improved it as suggested (Lines 137-139).
- (Line 162) the word "classify" is verb, and noun form is more appropriate in the sentence
resp: I have revised and improved it as suggested (Lines 170-171).
- (Line 170) I would assume that authors wanted to mention "test set" instead of "test rate"
resp: I have revised and improved it as suggested (Lines 177-178).
- (Line 248) I would like to discourage the usage of subjective word such as "remarkable" in this type of scientific article
resp: I have revised and improved it as suggested (Lines 291-292).
- (Line 280) I would like to discourage the usage of subjective word such as "perfect" in this type of scientific article
resp: I have revised and improved it as suggested (Lines 318-319).
Round 2
Reviewer 1 Report
Comments and Suggestions for Authors
My comments have been adequately addressed in the revision.
Author Response
Thank you for your valuable and insightful comments. This significantly contributed to the improvement of our manuscript.
Reviewer 3 Report
Comments and Suggestions for Authors
The authors seem to have significant amount of effort to improve the article in response to reviwers' requests.
Comments on the Quality of English LanguageWhile the revised version has some erros, those are acceptable.
Author Response
Thank you for your valuable and insightful comments. We appreciate your thorough review, and I have carefully revised the manuscript to address the identified typo errors.